# The structure of the endogenous ESX-3 secretion system

Nicole Poweleit[1,2], Nadine Czudnochowski[1,2], Rachel Nakagawa[1], Donovan D Trinidad[1,2], Kenan C Murphy[3], Christopher M Sassetti[3], Oren S Rosenberg[1,2]*

[1]Department of Medicine, Division of Infectious Diseases, University of California, San Francisco, San Francisco, United States; [2]Chan-Zuckerberg Biohub, University of California, San Francisco, San Francisco, United States; [3]Department of Microbiology and Physiological Systems, University of Massachusetts Medical School, Worcester, United States

**Abstract** The ESX (or Type VII) secretion systems are protein export systems in mycobacteria and many Gram-positive bacteria that mediate a broad range of functions including virulence, conjugation, and metabolic regulation. These systems translocate folded dimers of WXG100-superfamily protein substrates across the cytoplasmic membrane. We report the cryo-electron microscopy structure of an ESX-3 system, purified using an epitope tag inserted with recombineering into the chromosome of the model organism *Mycobacterium smegmatis*. The structure reveals a stacked architecture that extends above and below the inner membrane of the bacterium. The ESX-3 protomer complex is assembled from a single copy of the $EccB_3$, $EccC_3$, and $EccE_3$ and two copies of the $EccD_3$ protein. In the structure, the protomers form a stable dimer that is consistent with assembly into a larger oligomer. The ESX-3 structure provides a framework for further study of these important bacterial transporters.

*For correspondence: oren.rosenberg@ucsf.edu

## Introduction

Mycobacteria use a set of specialized secretion systems called ESX to transport proteins across their complex, diderm cell walls (*Gröschel et al., 2016*). Originally described as virulence factors in *Mycobacteria tuberculosis* (*Guinn et al., 2004*; *Hsu et al., 2003*; *Lewis et al., 2003*; *Stanley et al., 2003*), orthologs of ESX have since been discovered in most Gram-positive bacteria (*Bottai et al., 2017*), and are more generally referred to as Type VII secretion systems (*Bitter et al., 2009*). In mycobacteria there are five paralogous ESX operons (ESX 1–5) each of which encodes an inner membrane translocon complex consisting of three conserved Ecc proteins: EccB, EccC, and EccD. A fourth protein, EccE is conserved in all ESX operons except the ancestral ESX-4 operon and is also considered a part of the ESX translocon complex as it copurifies with EccB, EccC, and EccD (*Houben et al., 2012*). All Type VII secretion systems translocate proteins in the WXG100-superfamily, which share a common two-helix hairpin structure and are found as homo- or heterodimers (*Poulsen et al., 2014*) and are mutually dependent for secretion with other substrates (*Fortune et al., 2005*). In contrast to the general secretory apparatus (Sec), ESX substrates have been shown to be secreted in their folded, dimeric state (*Sysoeva et al., 2014*).

Structural and functional information has been reported for truncated and isolated, soluble domains of the ESX translocon complexes and their homologs (*Korotkova et al., 2015*; *Korotkova et al., 2014*; *Renshaw et al., 2005*; *Rosenberg et al., 2015*; *Strong et al., 2006*; *Wagner et al., 2016*; *Wagner et al., 2014*; *Wagner et al., 2013*; *Zhang et al., 2015*; *Zoltner et al., 2016*). A low resolution, negative stain electron microscopy structure of ESX-5 shows a translocon complex assembled into a hexamer (*Beckham et al., 2017*). Structures of other proteins encoded in

ESX operons including secreted substrates (*Ilghari et al., 2011*), substrate chaperons (*Ekiert and Cox, 2014*), and the protease MycP (*Solomonson et al., 2013*) have been solved. Despite revealing important functional information about ESX, structures of overexpressed and isolated proteins are insufficient to understand the regulated secretion of fully folded substrates. We therefore undertook structural studies of an endogenously expressed ESX-3 complex from the model organism *M. smegmatis* using cryo-electron microscopy (cryo-EM). During the preparation of this work for publication, a similar structure of the ESX-3 system expressed from a plasmid was published by another group (*Famelis et al., 2019*).

The ESX-3 translocon complex is important for iron acquisition (*Serafini et al., 2013*; *Siegrist et al., 2009*), cell survival (*Tinaztepe et al., 2016*), and virulence in pathogenic mycobacteria (*Tufariello et al., 2016*), and its role in iron homeostasis is conserved in the model system, *M. smegmatis* (*Siegrist et al., 2009*). The ESX-3 translocon complex proteins are transcribed in a single operon (*Li et al., 2017*), and expression of the ESX-3 operon is dependent on the transcriptional regulator IdeR, which controls iron metabolism (*Rodriguez et al., 2002*) and is required for growth in the human pathogen *M. tuberculosis* (*Pandey and Rodriguez, 2014*). The ESX-3 operon is 67% identical between the non-pathogenic model organism *M. smegmatis* and the pathogen *M. tuberculosis* over the 4354 amino acids of the ESX-3 operon. This high degree of conservation and essential role in cell growth makes ESX-3 an important candidate for small molecule inhibition (*Bottai et al., 2014*), as blockade of ESX-3 will both inhibit virulence in *M. tuberculosis* and kill a broad range of pathogenic mycobacteria.

## Results

A major innovation made possible by the dramatic improvements in cryo-EM (*Cheng, 2018*) is the ability to examine challenging protein samples at atomic resolution, even when samples are only available at low concentrations. When coupled with recent genetic manipulations that allow for facile insertion of chromosomal epitope and purification tags (*Murphy et al., 2018*), cryo-EM now holds the promise of routine structural characterization of many endogenously expressed protein complexes not previously tractable by structural techniques. We undertook the purification of the ESX-3 complex from the native host without the need for overexpression. To facilitate purification of the endogenous translocon complex, a cleavable EGFP tag was inserted into the chromosome of *M. smegmatis* mc(2)155 (wild type) and $\Delta ideR$ (*Dussurget et al., 1996*) strains at the C-terminus of $EccE_3$ via the ORBIT method (*Murphy et al., 2018*) (*Figure 1A*, *Figure 1—figure supplement 1*). $EccE_3$ is the final gene in the 11 gene long ESX-3 operon making insertion at this site less likely to disrupt regulation and expression of the operon. Deletion of the gene for the iron acquisition regulator IdeR greatly increases chromosomal expression of ESX-3 from negligible amounts of protein to a yield sufficient for purification and structure determination (*Figure 1—figure supplement 2A*). Components of ESX-3 were pulled down using an anti-GFP nanobody (*Rothbauer et al., 2008*) and the EGFP tag was proteolytically cleaved. After size exclusion chromatography, the peak fractions were pooled and analyzed biochemically and by cryo-EM .

### Global structure of the ESX-3 dimer

Four components of the ESX-3 translocon complex $EccB_3$, $EccC_3$, $EccD_3$ and $EccE_3$ were stably affinity-purified as a large molecular weight species of about 900 kDa (*Figure 1B and C*, *Figure 1—figure supplement 2B*). The sample was imaged by cryo-EM and reconstructed revealing a dimeric structure, which can be divided into four areas: the flexible periplasmic multimerization domain, the stable transmembrane region, the stable upper cytoplasmic region, and the flexible lower cytoplasmic motor domain (*Figure 1D*, *Table 1*). While the peak fraction does not contain particles of a larger size consistent with higher order oligomers, thorough examination of the void volume revealed a small number of particles in a higher oligomeric state (*Figure 1—figure supplement 3A–E*). The resolution of the ESX-3 dimer varies substantially in different parts of the electron microscopy map and this heterogeneity was partially resolved through data processing (*Figure 1—figure supplement 4* and *Figure 1—figure supplement 5*). Initially, the entire ESX-3 dimer was reconstructed to 4.0 Å resolution (*Figure 1—figure supplement 6A*). Using symmetry expansion, and focused classification and refinement techniques, the resolutions of targeted regions of the ESX-3 complex were improved to 3.7 Å for the transmembrane region and upper cytoplasmic region (*Figure 1—figure*

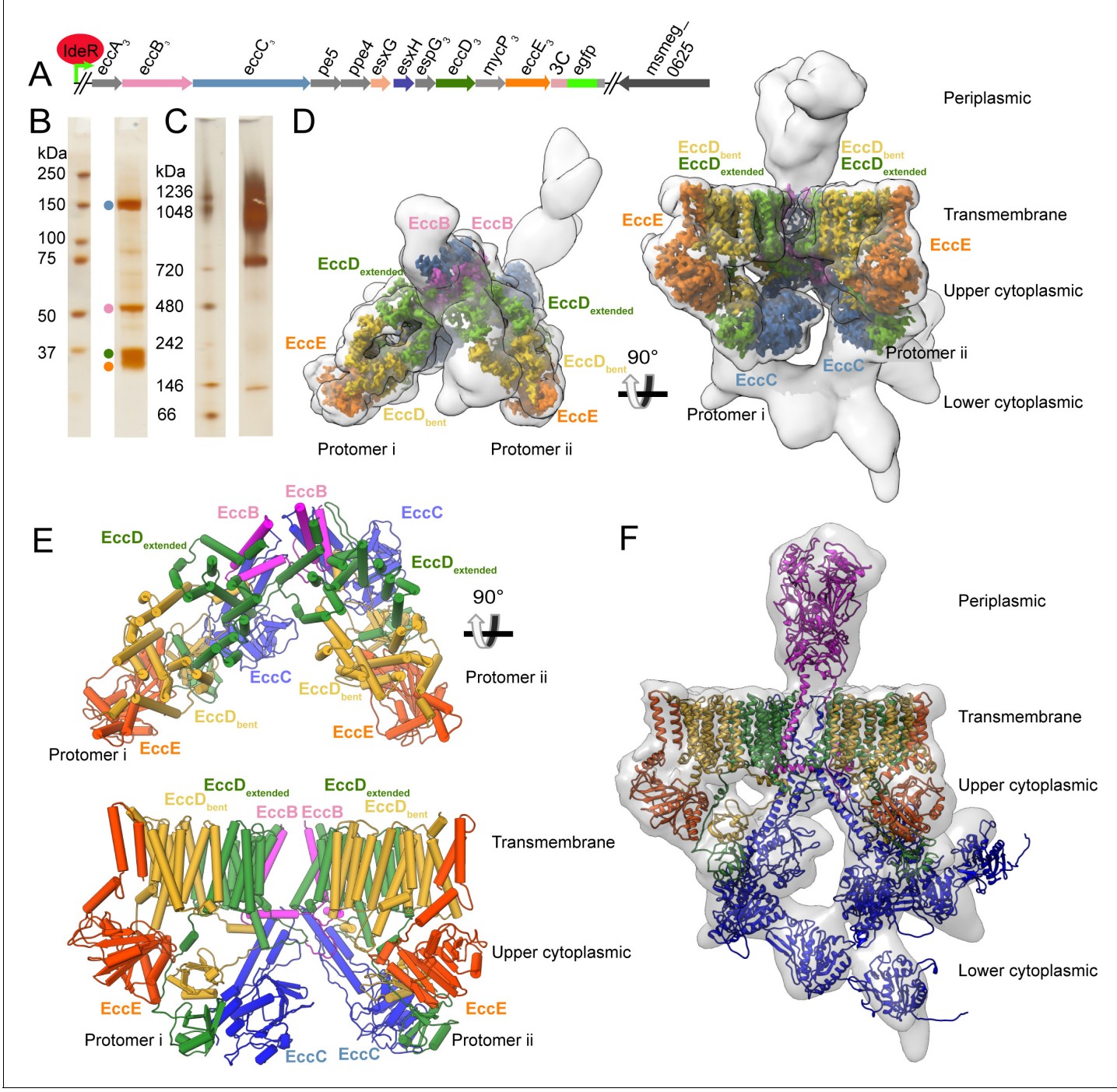

**Figure 1.** Overview of the ESX-3 tagging, purification, and structure. (**A**) The ESX-3 operon in *M. smegmatis* and the placement of the purification tag. Genomic deletion of *ideR* derepresses ESX-3 to boost expression for purification. (**B**) SDS-PAGE of purified ESX-3 shows four major bands corresponding to EccB$_3$, EccC$_3$, EccD$_3$, and EccE$_3$. (**C**) Blue native page of the purified ESX-3 complex shows a large molecular weight band around 900 kDa. (**D**) Merged maps of all focused refinement maps (gray transparency) of the ESX-3 dimer filtered to 10 Å resolution. The transmembrane and upper cytoplasmic focused maps (3.7 Å) are segmented by subunit showing one copy per protomer of EccB$_3$ (pink), EccC$_3$ (blue), EccE$_3$ (orange), EccD$_{3-bent}$ (yellow), and EccD$_{3-extended}$ (green). (**E**) Atomic models of the transmembrane and upper cytoplasmic regions. (**F**) A combined map of the full complex filtered to 10 Å resolution (gray transparency) with full models for each protein, EccD$_{3-bent}$ (yellow), EccD$_{3-extended}$ (green), EccE$_3$ (orange), EccC$_3$ (blue), and EccB$_3$ (pink).

The online version of this article includes the following figure supplement(s) for figure 1:

**Figure supplement 1.** ORBIT tagging of the chromosomal copy of EccE$_3$.

**Figure supplement 2.** ESX-3 dimer purification optimization.

*Figure 1 continued on next page*

*Figure 1 continued*

**Figure supplement 3.** Examination of the void volume.
**Figure supplement 4.** Initial data collection and initial model generation.
**Figure supplement 5.** Data processing workflow for final data collection.
**Figure supplement 6.** Consensus and focused refinements.

supplement 6B–D), 5.8 Å for the periplasmic region (*Figure 1—figure supplement 6E*), and ~7 Å resolution for the lower cytoplasmic region (*Figure 1—figure supplement 6F*). The highest resolution maps for each region were combined and filtered to the threshold of the lowest resolution map to form an overall 10 Å combination map for the entire ESX-3 dimer.

The ESX-3 dimer is comprised of ten total proteins, two copies each of $EccB_3$, $EccC_3$, and $EccE_3$ and four copies of $EccD_3$. Two pseudo-symmetric protomers referred to as i and ii, combine to form the ESX-3 dimer. Each protomer contains one copy of $EccB_3$, $EccC_3$, and $EccE_3$ and two conformationally distinct copies of $EccD_3$, referred to as $EccD_{3\text{-bent}}$ and $EccD_{3\text{-extended}}$ (*Figure 1E*) based on their highly asymmetric conformations. At 3.7 Å resolution, it was possible to build *de novo* atomic models for all observable amino acids in the transmembrane and upper cytoplasmic regions, except the two transmembrane helices of $EccC_3$ (*Figure 1E* and *Supplementary file 1*). The lower

**Table 1.** Data collection and refinement statistics.
Collection parameters for initial test data set and void peak analysis on the Talos Arctica and final collection on Titan Krios microscopes. Refinement details for initial model, consensus map, and focused refinements.

**Data Collection**

| Collection | Initial Screening | Collection 1 | Collection 2 | Void peak |
|---|---|---|---|---|
| Microscope | Talos Arctica | Titan Krios | Titan Krios | Talos Arctica |
| Voltage (kV) | 200 | 300 | 300 | 200 |
| Detector | Gatan K2 | Gatan K2 | Gatan K2 | Gatan K3 |
| Pixel size (Å/pixel) | 1.14 | 0.82 | 0.82 | 0.9 |
| Exposure Time (s) | 9 | 10 | 10 | 11.7 |
| Electron dose (e-/Å$^2$) | 63 | 80 | 67 | 58 |
| Defocus range (μm) | 1.5-2.5 | 0.4-1.2 | 0.6-1.4 | 1.5-2.5 |
| Number of micrographs | 2,499 | 2,705 | 4,632 | 1,215 |

**Consensus Reconstruction**

| Data Set | Initial Screening | Collection 1 & 2 | | Void peak |
|---|---|---|---|---|
| Software | Relion 2.1, cisTEM, and cryosparc | Relion 3.0 | | cisTEM |
| # of particles, picked | 240,000 | 778,149 | | 259,333 |
| # of particles post, Class2D | 138,000 | 554,901 | | 640 |
| # of particles post, Class3D | 46,830 | 362,438 | | NA |
| # of particles post, skip align Class3D | NA | 90,479 | | NA |
| Symmetry | C1 | C1 | | NA |
| Map sharpening B-factor (Å$^2$) | NA | -160 | | NA |
| Final resolution (Å) | 4.7 Å | 4.0 Å | | NA |

**Focused Refinements**

| Location of focus | # of particles | Resolution |
|---|---|---|
| Protomer i | 76,967 | 3.8 |
| Protomer ii | 90,479 | 3.8 |
| Symmetry expanded protomer | 52,067 | 3.7 |
| Periplasmic EccB3 | 70,000 | 5.8 |
| ATPase 1, 2, and 3 | 30,000 | 7 |

resolution regions of density, the EccC$_3$ transmembrane helices, EccC$_3$ ATPase 1, 2, and 3 domains and the EccB$_3$ periplasmic domain, were flexibly fit using homology models. Using this hybrid approach, a model of the entire ESX-3 dimer has been produced (*Figure 1F*).

## EccD$_3$ forms a homodimer that encloses a large hydrophobic cavity

There are two copies of EccD$_3$ in each ESX-3 protomer (*Figure 2A*). The ubiquitin-like N-terminal domain of each EccD$_3$ molecule interacts with EccE$_3$ and EccC$_3$ in the cytoplasm, and a long linker joins the soluble domain of EccD$_3$ to 11 transmembrane helices (*Figure 2B* and *Figure 2—figure supplement 1A–F*). The four EccD$_3$ molecules account for 44 of the total 54 transmembrane helices observed in the ESX-3 dimer. A distinct transmembrane cavity is formed by dimerization of the two copies of EccD$_3$ in each protomer with a cross-sectional diameter of ~20×30 Å without significant regions of constriction. Transmembrane helices 1, 9 and 10 interact across the cavity dimer interface in a tight bundle making passive lipid transport into the membrane from the cavity unlikely (*Figure 2C*). The inner surface of the periplasmic half of the cavity is composed primarily of hydrophobic residues and in our maps, eight extended densities consistent with hydrophobic lipid tails or detergent molecules line the periplasmic inner face of the cavity (*Figure 2C*). In contrast, the

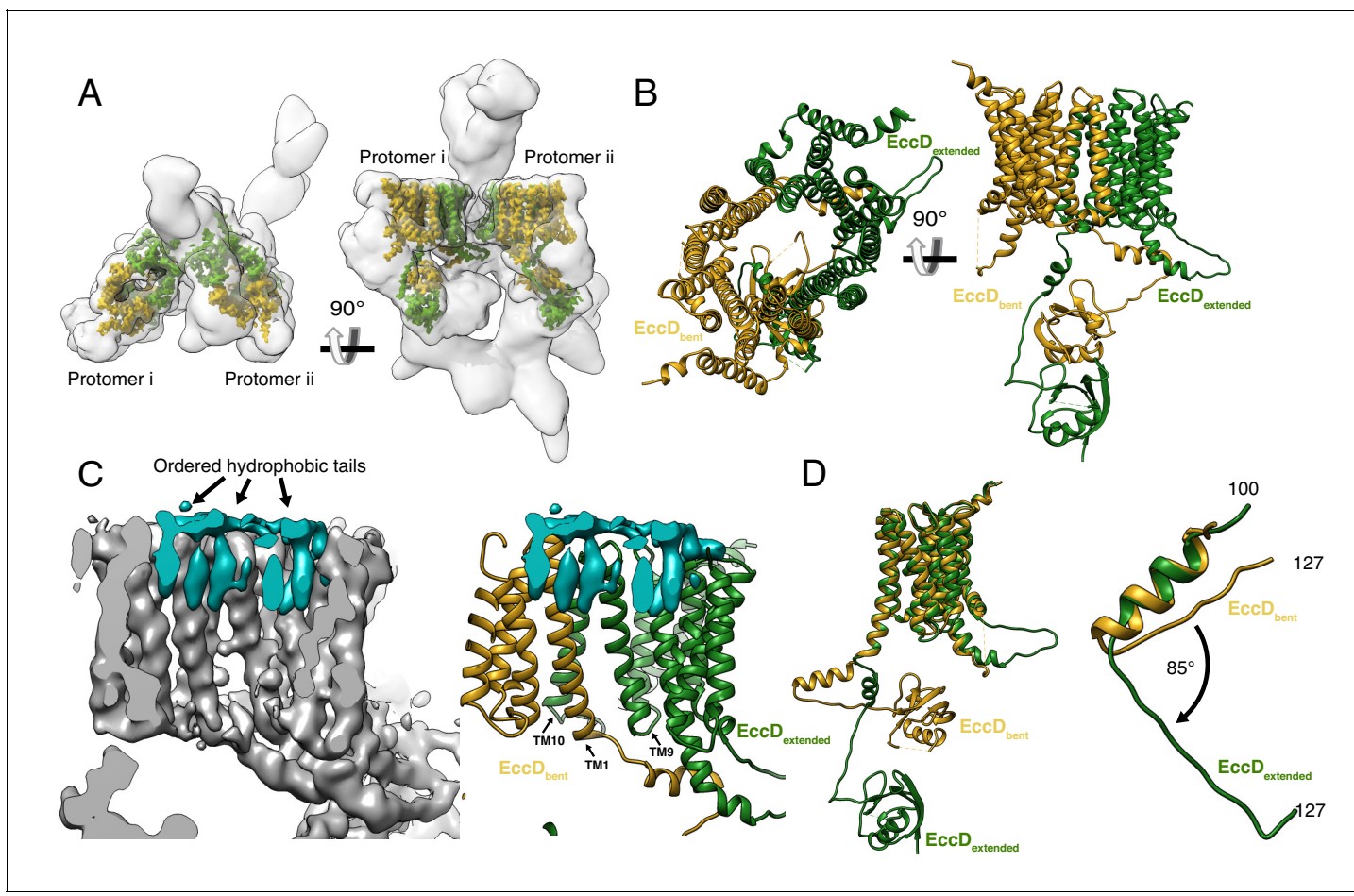

**Figure 2.** The structure of EccD$_3$. (A) EccD$_{3\text{-bent}}$ (yellow) and EccD$_{3\text{-extended}}$ (green) in the context of the overall ESX-3 dimer (gray transparency). (B) Atomic models of EccD$_{3\text{-bent}}$ and EccD$_{3\text{-extended}}$ (C) An unsharpened electron microscopy density map of the ESX-3 dimer shows extra densities consistent with lipid or detergent molecules (teal) on the periplasmic face of the EccD$_3$ cavity. (D) EccD$_{3\text{-bent}}$ (yellow) and EccD$_{3\text{-extended}}$ (green) aligned based on the transmembrane regions shows two distinct conformations of the EccD$_3$ cytoplasmic domains. Amino acids 100–127 of EccD$_3$ adopt a bent (yellow) and an extended (green) conformation.

The online version of this article includes the following figure supplement(s) for figure 2:

**Figure supplement 1.** EccD$_3$ map and model.

cytoplasmic face of the cavity has several polar residues and ordered hydrophobic densities are not visible.

In the cytoplasm below the membrane, a stable, upper cytoplasmic region is formed by interactions between the soluble domains of $EccD_{3-bent}$, $EccD_{3-extended}$, $EccE_3$, and $EccC_3$. The linker joining the cytoplasmic ubiquitin-like domain to transmembrane helix 1 (residues 100–127) of $EccD_3$ conserves a high sequence identity throughout evolution (*Ashkenazy et al., 2016*), yet it adopts two distinct secondary structures resulting in the asymmetric placement of the cytoplasmic domains of $EccD_3$ (*Figure 2D* and *Figure 2—figure supplement 1A–G*). In $EccD_{3-bent}$, residues 100–127 are bent, folding into an α-helix and forming a nexus of stabilizing contacts with $EccB_3$ and $EccC_3$ (*Figure 2—figure supplement 1H*). In $EccD_{3-extended}$, residues 100–127 are extended and fold into a shorter α-helix that interacts with $EccE_3$ and the cytoplasmic domain of $EccD_{3-bent}$ (*Figure 2—figure supplement 1I*). This conformational flexibility suggests that if residues 100–127 were released from their associations with $EccC_3$ and $EccE_3$ they could rearrange into the alternative bent or extended conformation with little energetic barrier.

## $EccC_3$ and $EccE_3$ make extensive, stabilizing interactions with the asymmetric, cytoplasmic domains of $EccD_3$

The next component of the stable upper cytoplasmic region is $EccE_3$. $EccE_3$ is positioned at the front of the ESX-3 dimer (*Figure 3A*), where the conserved transmembrane helix 1 of $EccE_3$ interacts with helix 11 of $EccD_{3-bent}$ in the membrane. Helix 1 is followed by a second $EccE_3$ transmembrane helix, a linker helix, and then extends into the cytoplasm (*Figure 3B and C* and *Figure 3—figure supplement 1A–D*). The anti-parallel β-sheets of the cytoplasmic domain of $EccE_3$ have weak structural homology to glycosyl transferase proteins, however, the nucleotide binding pocket is absent in $EccE_3$, leaving it incapable of performing this function (*Figure 3—figure supplement 1E and F*, *Supplementary file 2*). $EccE_3$ does not have another obvious ligand or catalytic site. Two conserved helices in the cytoplasmic region of $EccE_3$ between amino acids 133 and 163 form extensive stabilizing interactions with both subunits of $EccD_3$ (*Figure 3D*, *Figure 3—figure supplement 1G*). These interactions hold the flexible linker of $EccD_{3-extended}$ in the extended conformation and sterically hinder $EccD_{3-extended}$ from assuming the bent conformation (*Figure 3—figure supplement 1H*). $EccE_3$ does not form direct protein-protein interactions with either $EccB_3$ or $EccC_3$ suggesting the contacts with $EccD_{3-extended}$ and $EccD_{3-bent}$ are extremely stable as $EccE_3$ was the tagged protein used to immunoprecipitate the ESX-3 dimer.

The final component of the stable upper cytoplasmic region is the domain of unknown function (DUF) of $EccC_3$. $EccC_3$ extends from the membrane into the upper and lower cytoplasmic regions (*Figure 4A*). Amino acids 1–33 and 94–403 of $EccC_3$ were built *de novo* into the higher resolution region of the electron microscopy map revealing the structure of the DUF domain (*Figure 4B*, *Figure 4—figure supplement 1A–C*). The *de novo* model of the DUF has the typical Rossman fold of a nucleotide hydrolysis domain (*Figure 4—figure supplement 1D and E*) and its closest homolog by Dali search is the ATPase 1 domain of EccC of *T. curvata*. It is linked to the transmembrane domains by a long helical bundle making extensive contacts with the flexible linker region of $EccD_{3-bent}$ (*Figure 4C*). The DUF makes additional stabilizing contacts with the ubiquitin-like domains of $EccD_{3-bent}$ and $EccD_{3-extended}$ in the cytoplasm (*Figure 4D*).

When the transmembrane and upper cytoplasmic regions are compared between protomers, only the transmembrane helices of $EccC_3$ and the N-terminal tail of $EccB_3$ differ (*Figure 4—figure supplement 2A and B*), otherwise the protomers are superimposable. All four $EccC_3$ transmembrane helices were modeled at 6 Å resolution through a combination of homology modeling and molecular dynamics. In protomer i, transmembrane helix 2 forms lipid mediated hydrophobic interactions with the transmembrane helix of $EccB_3$ in protomer i, and transmembrane helix 1 interacts with transmembrane helix 2 of $EccC_3$ in protomer ii. Transmembrane helix 1 of $EccC_3$ in protomer ii is shifted relative to the protomer i conformation and does not directly interact with other proteins.

## The $EccC_3$ motor domains are flexible and asymmetric across the dimer

The motor domains containing the $EccC_3$ ATPase 1, 2 and 3 hang below the DUF domain in the flexible lower cytoplasmic region. They were resolved at a lower resolution than the upper cytoplasmic domain, but they are clearly asymmetric between protomers i and ii (*Figure 4A*). Although the

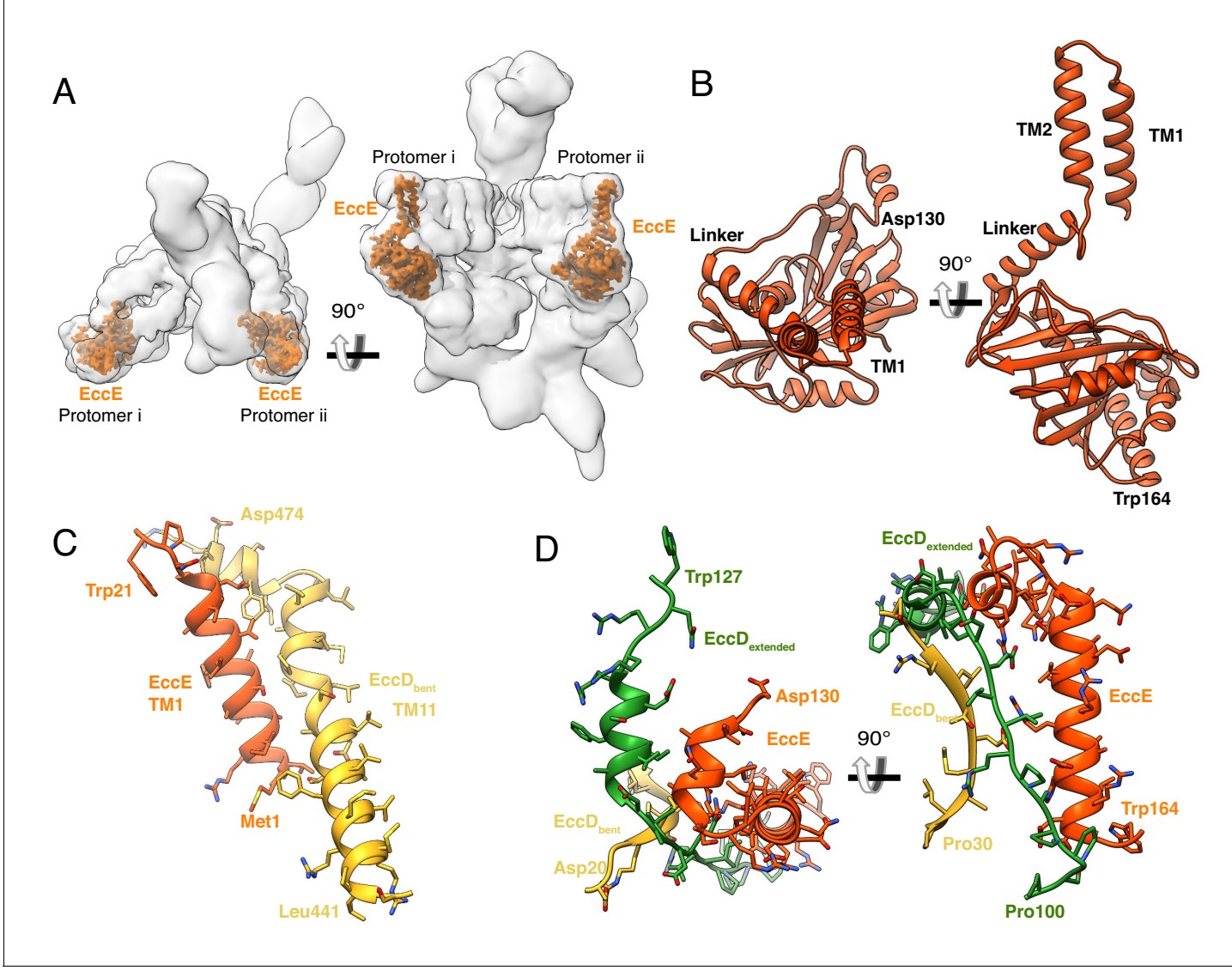

**Figure 3.** The structure and protein-protein interactions of EccE$_3$. (**A**) The placement of EccE$_3$ in the overall ESX-3 dimer. (**B**) Atomic model of EccE$_3$ (**C**) Transmembrane helix 1 of EccE$_3$ interacts with transmembrane helix 11 of EccD$_{3\text{-bent}}$ (**D**) Two soluble helices of EccE$_3$ interact with EccD$_{3\text{-extended}}$ and EccD$_{3\text{-bent}}$.

The online version of this article includes the following figure supplement(s) for figure 3:

**Figure supplement 1.** EccE$_3$ map and model.

EccC$_3$ ATPase 1 domains in both protomers are in a similar location relative to the DUF, the ATPase 2 and 3 domains do not superimpose across protomers even at low resolution (*Figure 4—figure supplement 2C*) suggesting significant asymmetry between these domains. In protomer i, a homology model based on existing EccC structures fits well into the density; however in protomer ii, the interface between ATPase 1 and 2 is broken relative to the crystal structure with ATPase 2 and ATPase 3 rotated away from the crystal structure interface.

## EccB$_3$ extends into the periplasm and stabilizes dimer formation

The ESX-3 dimer is stabilized by cross-protomer interactions formed by the two EccB$_3$ proteins. EccB$_3$ begins in the cytoplasm with a flexible N-terminal tail leading into a linker helix, followed by a single-pass transmembrane helix, and an extended periplasmic domain (*Figure 5A and B*, *Figure 5—figure supplement 1A and B*). The N-terminal tail of EccB$_3$ from protomer i forms extensive cross-

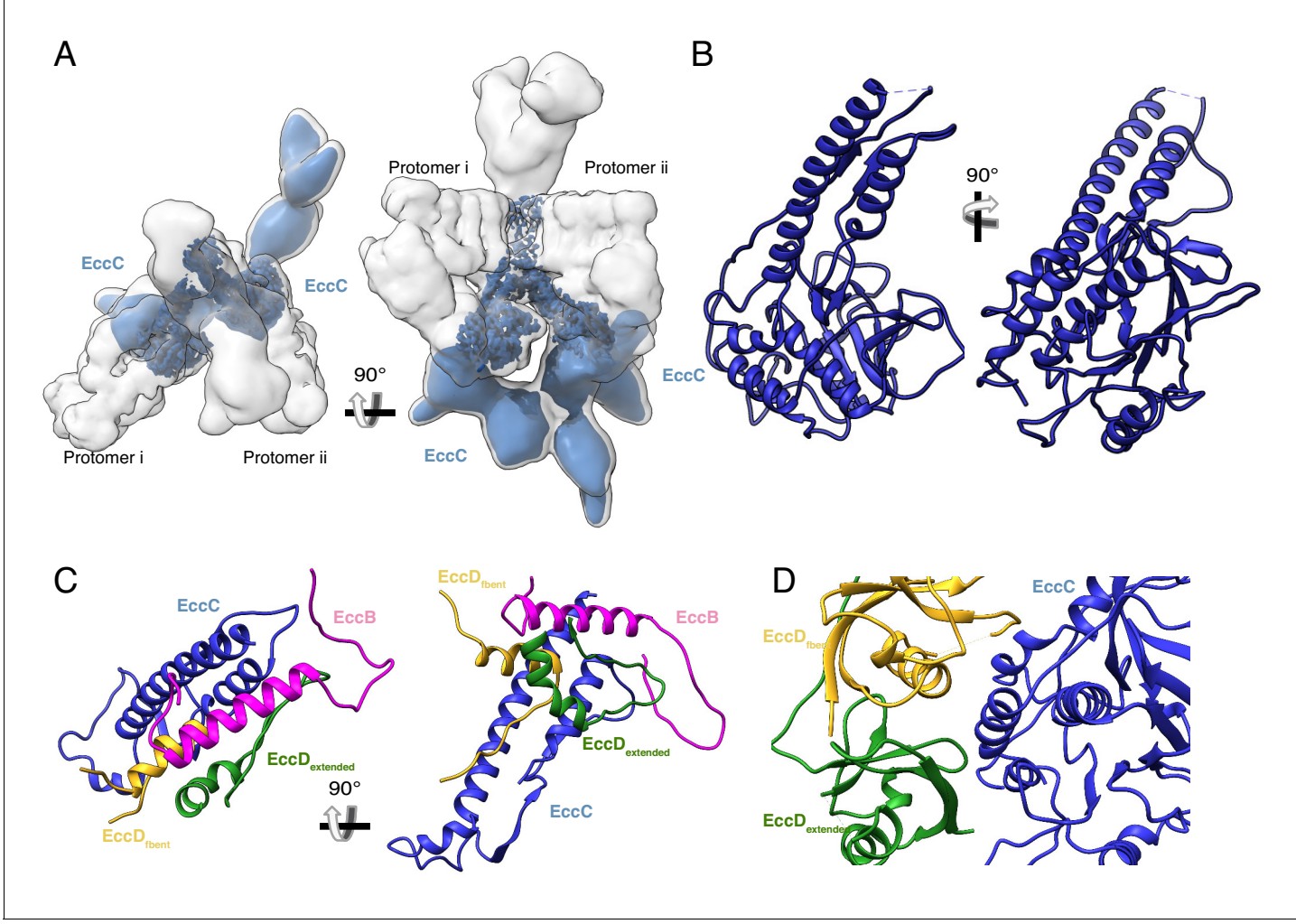

**Figure 4.** The structure and protein-protein interactions of $EccC_3$. (A) The placement of $EccC_3$ in the overall ESX-3 dimer. (B) Atomic model of the $EccC_3$ DUF (C) The stalk helices of $EccC_3$ interact with $EccB_3$, $EccD_{3-bent}$, and $EccD_{3-extended}$ (D) Interactions between $EccC_3$ and the ubiquitin-like domains of $EccD_{3-bent}$ and $EccD_{3-extended}$ in the cytoplasm.

The online version of this article includes the following figure supplement(s) for figure 4:

**Figure supplement 1.** $EccC_3$ map and model.

**Figure supplement 2.** Conformational differences between protomer i and protomer ii.

protomer contacts with $EccB_3$, $EccC_3$, $EccD_{3-bent}$ and $EccD_{3-extended}$ from protomer ii (*Figure 5C*, *Supplementary file 3*). The linker helix of $EccB_3$ forms further protein-protein interactions with $EccC_3$ and $EccD_{3-bent}$. The transmembrane helix of $EccB_3$ interacts with transmembrane helix 11 of $EccD_{3-extended}$. Two hydrophobic tails consistent with a lipid or detergent molecules link the transmembrane helix of $EccB_3$ to transmembrane helix 2 of $EccC_3$ (*Figure 5D*). The two $EccB_3$ periplasmic domains share a large interaction interface across the protomers further stabilizing dimerization. Homology models of two $EccB_3$ proteins can be docked into the periplasmic domain (*Figure 5—figure supplement 1C*); however, this region is not resolved sufficiently to identify specific interactions. The majority of cross-protomer interactions involve $EccB_3$, suggesting the periplasmic domain is essential for oligomerization.

## A hexameric model of ESX-3

Previous reports have shown ESX-1 and ESX-5 form hexamers or higher order multimers (*Beckham et al., 2017*; *Houben et al., 2012*). We modeled a higher order oligomeric state of ESX-3 based on the low-resolution negative stain structure of ESX-5, which had C6 symmetry imposed

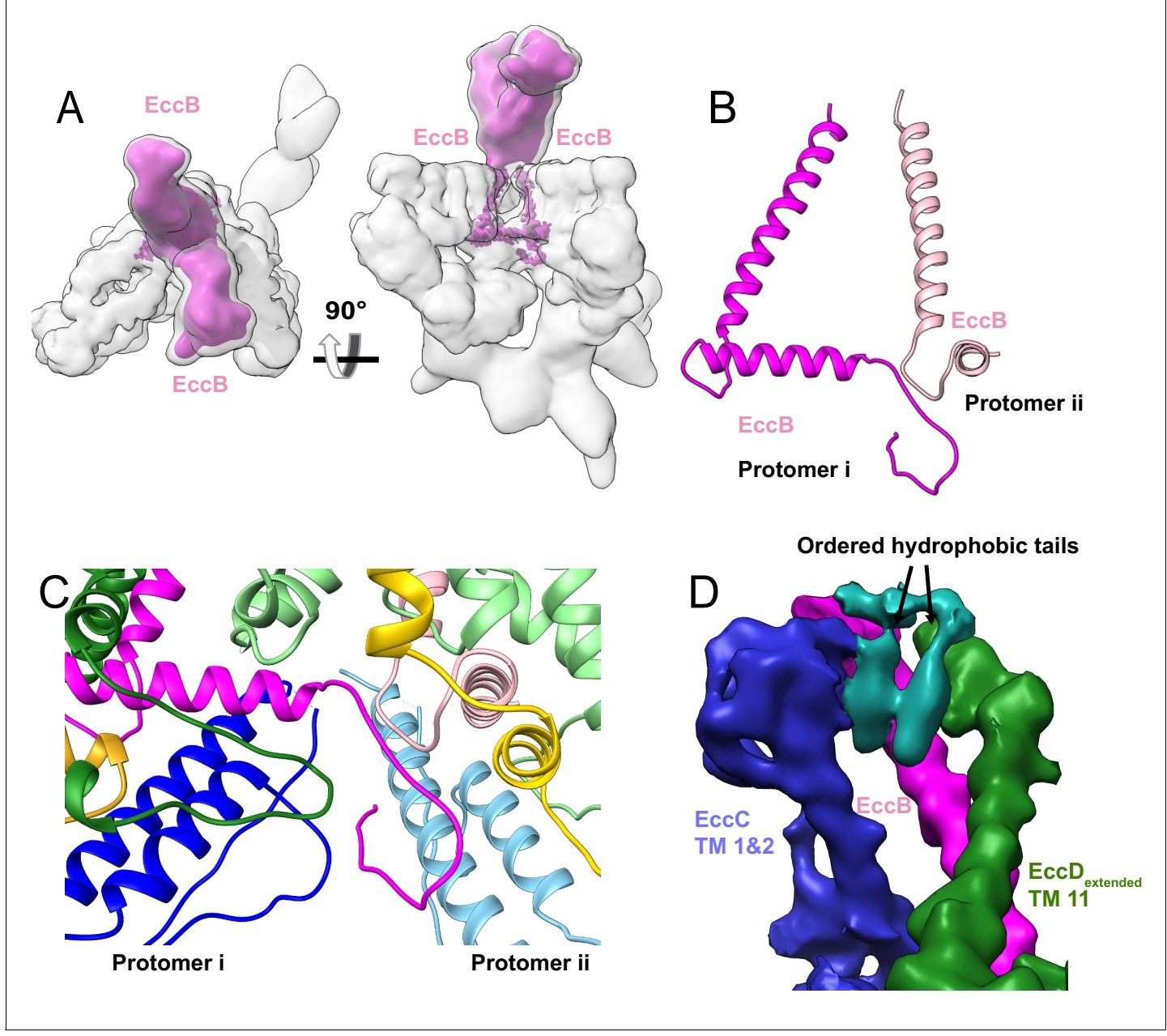

**Figure 5.** The periplasmic multimerization domain. (A) EccB$_3$ (pink) in the context of the overall ESX-3 dimer (gray transparency). EccB$_3$ has a single-pass transmembrane domain which extends into a large periplasmic domain which was resolved at 5.8 Å resolution. (B) Atomic models of the EccB$_3$ cytoplasmic and transmembrane domains, amino acids 14–93 and 32–93. (C) The N-terminus of EccB$_3$ forms extensive cross-protomer contacts with EccC$_3$ (blue), EccD$_{3-bent}$ (yellow), and EccD$_{3-extended}$ (green). (D) An unsharpened map of the ESX-3 dimer reveals ordered densities consistent with lipids or detergent molecules mediating the interactions between the EccB$_3$ transmembrane helix (marked with a pink dot) and the EccC$_3$ transmembrane helices.

The online version of this article includes the following figure supplement(s) for figure 5:

**Figure supplement 1.** EccB$_3$ maps and models.

during refinement (EMDB 3596). The ESX-3 translocon complex was modeled as a trimer of dimers by low pass filtering the density to 6 Å and docking into the ESX-5 negative stain map (*Figure 6— figure supplement 1A*). The model of the ESX-3 dimer transmembrane and upper cytoplasmic regions, including the EccC$_3$ transmembrane helices modeled to 6 Å resolution, was fitted into the trimerically positioned ESX-3 density maps (*Figure 6—figure supplement 1B*). The angle between

protomers in the trimer of dimers model alternates between 72° (the angle between protomers i and ii in the ESX-3 dimer map) and 48° and contains both experimentally observed conformations of $EccC_3$. The complete model of the ESX-3 translocon complex was docked into the ESX-5 negative stain map in the same manner revealing major clashes between the low resolution periplasmic and motor domains in a hexameric form (*Figure 6—figure supplement 1C*). Accommodation of a hexameric complex would require extensive rearrangement of both $EccC_3$ and $EccB_3$.

## Discussion

The ESX-3 structure presented here is purified without the addition of substrates or nucleotide. It is therefore likely to be in a conformation representing the end of the translocation cycle, awaiting either the direct binding of substrates, the binding of nucleotide or both, to reset a substrate-binding competent state. By fitting our dimer structure into a prior low resolution envelope we suggest a model of the oligomeric state of the complex, in close agreement with Famelis et al. However, even allowing for major rearrangements in the more flexible regions of $EccC_3$ and $EccB_3$, a model built by the static trimerization of the experimentally determined dimer structure cannot itself explain the mechanism of action of ESX-3 secretion. The existence of an R-finger catalytic site for ATPase 1 of $EccC_3$ (*Rosenberg et al., 2015*) requires the R-finger of one protomer to insert into the active site of another protomer. Given the ~65 Å distance we observe between ATPase 1 domains, the completion and activation of the catalytic site of ATPase 1 by an R-finger will necessitate an extremely large rearrangement of the position of the ATPase domains. How might this rearrangement occur? We propose movements in the highly flexible $EccD_3$ linker lead to the release of $EccE_3$ and $EccC_3$ from their rigid positions, thus allowing for a rearrangement of the ATPase domains into an active conformation (*Figure 6*).

Once $EccC_3$ assembles into an oligomeric state, the substrate proteins will need to translocate through the inner membrane. We have considered two models for how pore formation and transit might occur: 1) through a pore created by the oligomerization of $EccC_3$ and $EccB_3$ or 2) through the large cavity created by the dimerization of $EccD_3$. In the first model (*Figure 6A*), the resting state of an ESX translocon complex is a hexamer, with disordered $EccC_3$ ATPase domains free in the cytosol, stabilized by interactions with proteins not seen in the structures presented here (*van Winden et al., 2016*). It is possible the rare multi-dimer oligomeric state we see in the void volume, and also seen by Famelis et al., represents this state. In a hexamer model, the center of the multimer is formed by the transmembrane helices of $EccC_3$ and $EccB_3$, which create a cavity that could serve as a pore for translocation of substrates. These transmembrane helices are largely hydrophobic and do not contain obvious residues that would allow for the conductance of hydrated substrates. Thus the production of a protein transit channel would require either a large, conformational change in the transmembrane helices, likely facilitated by movements in the cytoplasmic domains of $EccC_3$, $EccD_3$ and $EccE_3$, or a novel mechanism of action for transit through the central pore.

A hexameric pore created by $EccC_3$ agrees well with the documented mechanism of action for motor ATPases in the additional strand catalytic E (ASCE) division of P-loop NTPases (*Erzberger and Berger, 2006*), which includes $EccC_3$. A hexameric pore also agrees with the proposed mechanism of action for other bacterial secretion systems, such as the Type IV secretion system VirD4 coupling protein (*Gomis-Rüth et al., 2001*; *Hormaeche et al., 2002*), which is related evolutionarily to $EccC_3$ (*Iyer et al., 2004*). The hexamer model is thus firmly grounded in the motor ATPase and bacterial secretion systems literature, although the oligomeric state of VirD4 has recently been called into question (*Redzej et al., 2017*) and remains controversial (*Llosa and Alkorta, 2017*).

In a second, more speculative model, $EccD_3$ dimers form a channel for translocation of substrates (*Figure 6B*). The large cavity found in the $EccD_3$ dimer is striking and by structural homology, is unlike any other membrane protein in the Protein Data Bank. In our density maps, the $EccD_3$ dimer cavity appears capped on the periplasmic side by a dense layer of lipids. In contrast, on the cytoplasmic side the cavity does not exhibit bound lipids due to the polar residues lining the lower half of the cavity. The large cavity is of sufficient diameter to transit a folded EsxG/H dimer, however given the strong hydrophobicity of the cavity the mechanism would not be mediated by water and would require a novel mechanism of secretion that has not been seen in other bacterial secretion systems. It is also possible that the cavity exists to transit a non-protein substrate such as a specific

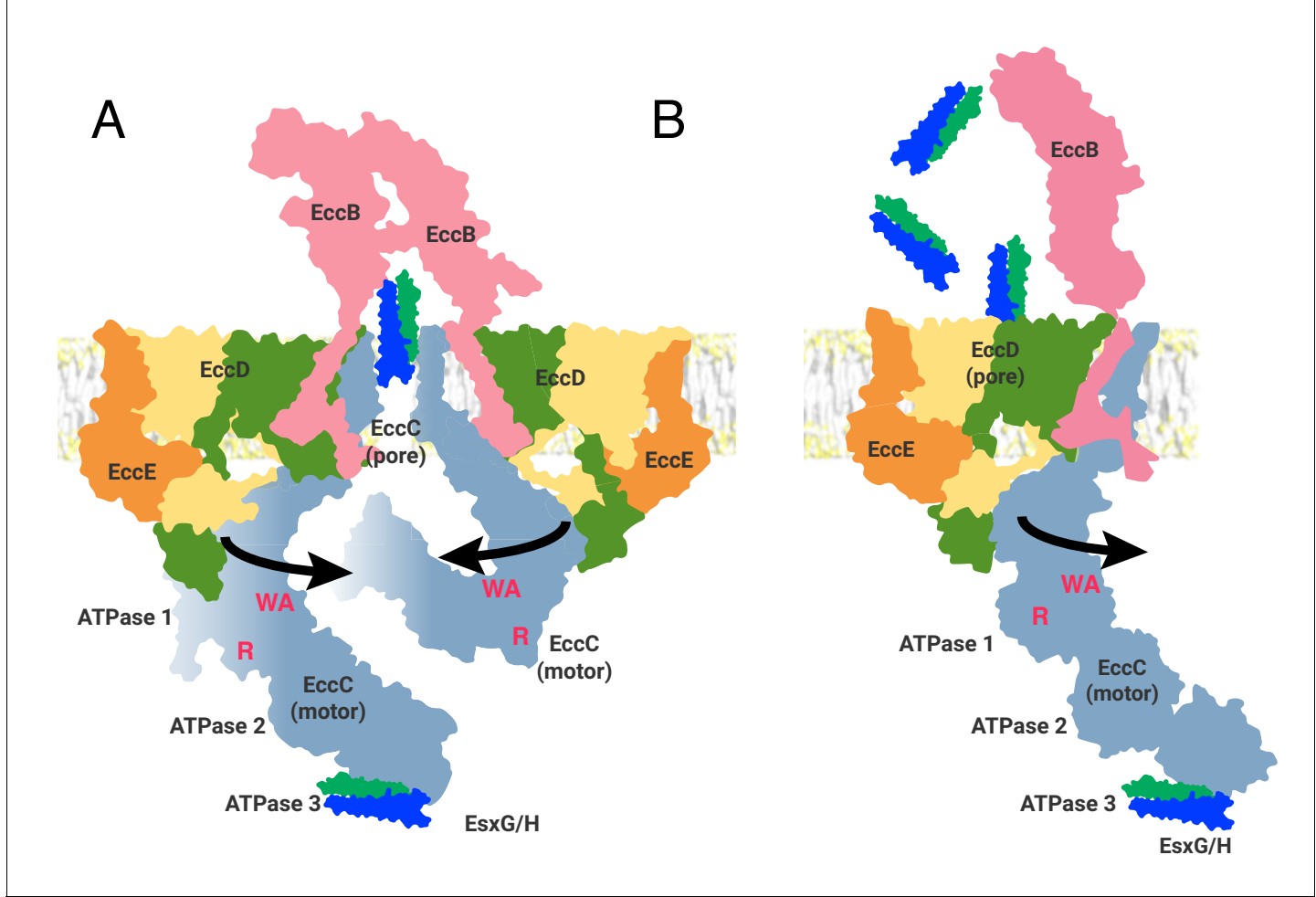

**Figure 6.** Two models of the ESX-3 translocon complex. ATPase activity entails, at a minimum, oligomerization of ATPase 1 to bring the R-finger (R) into proximity of the catalytic site, marked by the Walker A motif (WA). This requires at least 65 Å of movement from the position seen in the structure. (A) The first model of ESX substrate secretion involves trimerization of the ESX-3 dimer followed by multimerization of the EccC ATPase domains into a stack of one to four rings of ATPases (B) The second model shown through the function of a single protomer of the ESX-3 complex. Substrates are selected by interaction with ATPase 3 of EccC and transported via the upper cytoplasmic region to the EccD cavity for secretion. The online version of this article includes the following figure supplement(s) for figure 6:

**Figure supplement 1.** A hexameric model of the ESX-3 dimer.

mycobacterial lipid. The ability to transport non-protein substrates could resolve some of the mysteries that remain about the relationships between ESX systems, cell wall stability, lipid content, and nutrient acquisition (*Barczak et al., 2017*; *Bosserman and Champion, 2017*; *Siegrist et al., 2014*; *Tufariello et al., 2016*).

As each protomer contains an $EccD_3$ cavity, the second model, proposing translocation through $EccD_3$, does not require hexamerization. However, this model is also not incompatible with hexamerization, which would not block a substrate path through $EccD_3$. Further, the role of the hexamer may not be to form a central channel for substrate transit. Rather, hexamerization could serve some other purpose. For example, it may tether functional dimers together, facilitate localization, or increase local concentration and allosteric control of enzymatic activity (*Kuriyan and Eisenberg, 2007*).

Although the ESX-3 structure presented here allows for mechanistic hypotheses about the transit of substrates across the inner-membrane, it does not provide sufficient information to allow for a structural model of transit across the outer mycomembrane. The $EccB_3$ periplasmic domain (*Wagner et al., 2016*) has been found to have similarity to the peptidoglycan binding phage protein

PlyCB, which forms a ring inside the bacterial cell wall facilitating phage entrance into the cell. A hypothesis is that EccB$_3$ is anchored to a larger outer membrane complex, but the purification conditions we have employed remove proteins required for its stabilization.

The description of ESX-3 presented here agrees in both protein composition and structure with the work recently published by Famelis et al. Given the high sequence conservation among the ESX systems in mycobacteria and related actinobacteria, these structures likely represent an excellent framework for the structural modeling of the other ESX systems. Together, these structures provide a wealth of information about protein-protein interaction interfaces and ESX complex architecture, which can be used to guide structure-based drug design and to generate hypotheses for further mechanistic investigations.

# Materials and methods

## Key resources table

| Reagent type (species) or resource | Designation | Source or reference | Identifiers | Additional Information |
|---|---|---|---|---|
| Biological sample Mycobacterium smegmatis | *mc(2)155* | ATCC | 700084 | Wild type strain |
| Biological sample Mycobacterium smegmatis | *mc(2)155 with ideR::mγδ200 (KanR)* | *Dussurget et al., 1996*, Provided by GM Rodriguez | | |
| Biological sample Mycobacterium smegmatis | *mc(2)155, MSMEG_0626-3C-EGFP* | This paper | | |
| Biological sample Mycobacterium smegmatis | *mc(2)155 with ideR::mγδ200 (KanR), MSMEG_0626-3C-EGFP* | This paper | | |
| Recombinant DNA reagent | pKM444 | *Murphy et al., 2018* | Addgene Plasmid #108319 | Plasmid encoding for Che9c phage RecT and Bxb1 phage Integrase |
| Recombinant DNA reagent | pKM444 - zeo | This paper | | Addition of zeocin resistance cassette to pKM444 plasmid |
| Recombinant DNA reagent | pKM468-3C-EGFP | This paper | | Modified ORBIT tagging plasmid |
| Sequenced-based reagent | ORBIT targeting oligonucleotide | This paper | Oligo | 5' TGTGCGTTCCACTGGTTCCC CGGCAACCACCTGCTGCACGTG AGCCAGCCGGACTACCTAGGTT TGTACCGTACACCACTGAGACC GCGGTGGTTGACCAGACAAACC CGCCGGATGACCCGCTTCCTGC GCGGCTTCATGTTCGACTGAAC CCTTCACCGAGGTCCG 3' |
| Antibody | Anti-GFP stabilized antibody preparation | Roche | Cat. #: 11814460001 | Dilution 1:10000 |
| Antibody | Anti-mouse IgG HRP-conjugated antibody | R&D Systems | Cat. #: HAF007 | Dilution 1:5000 |
| Antibody | Goat anti-rabbit IgG antibody HRP | GenScript | Cat. #: A00098 | Dilution 1:5000 |
| Antibody | Rabbit anti-GroEL | Sigma-Aldrich | Cat. #: G6532-.5ML | Dilution 1:5000 |
| Peptide, recombinant protein | Human Rhinovirus (HRV) 3C Protease | Thermo Scientific Pierce | Cat. #: 88946 | |

*Continued on next page*

*Continued*

| Reagent type (species) or resource | Designation | Source or reference | Identifiers | Additional Information |
|---|---|---|---|---|
| Commercial assay or kit | Superose 6 Increase 10/300 GL | GE Healthcare | Cat. #: 29091596 | |
| Chemical compound, detergent | DDM, n-Dodecyl-β-D-Maltopyranoside | Inalco | Cat. #: 1758-1350 | |
| Chemical compound, detergent | GDN, glyco-diosgenin | Anatrace | Cat. #: GDN101 | |
| Other | NativePAGE 3-12% Bis-Tris Protein Gels, 1.0 mm, 10-well | ThermoFisher Scientific | Cat. #: BN1001BOX | |
| Commercial assay or kit | Pierce Silver Stain Kit | Pierce | Cat. #: PI24612 | |
| Software, algorithm | SerialEM | *Mastronarde, 2005* | | https://bio3d.colorado.edu/SerialEM/ |
| Software, algorithm | MotionCor2 | *Zheng et al., 2017* | | http://msg.ucsf.edu/em/software/motioncor2.html |
| Software, algorithm | CTFfind4 | *Rohou and Grigorieff, 2015* | | https://grigoriefflab.umassmed.edu/ctffind4 |
| Software, algorithm | RELION | *Scheres, 2012* | | cam.ac.uk/relion/index.php/Download_%26_install |
| Software, algorithm | cisTEM | *Grant et al., 2018* | | https://cistem.org/ |
| Software, algorithm | cryosparc | *Punjani et al., 2017* | | https://cryosparc.com/docs/reference/install/ |
| Software, algorithm | pyem | *Asarnow et al., 2019* | | https://github.com/asarnow/pyem |
| Software, algorithm | coot | *Emsley et al., 2010* | | https://www2.mrc-lmb.cam. |
| Software, algorithm | raptorX | *Källberg et al., 2012* | | http://raptorx.uchicago.edu/ |
| Software, algorithm | phenix real space refine | *Afonine et al., 2018* | | https://www.phenix-online. |
| Software, algorithm | MDFF | *Trabuco et al., 2009* | | org/documentation/reference/refinement.html |
| Software, algorithm | namdinator | *Kidmose et al., 2019* | | https://namdinator.au.dk/ |
| Software, algorithm | pisa | *Krissinel, 2015* | | http://www.ccp4.ac.uk/pisa/ |
| Software, algorithm | chimera | *Pettersen et al., 2004* | | https://www.cgl.ucsf.edu/chimera/ |
| Software, algorithm | chimeraX | *Goddard et al., 2018* | | https://www.cgl.ucsf.edu/chimerax/ |
| Software, algorithm | DALI | *Holm and Laakso, 2016* | | http://ekhidna2.biocenter.helsinki.fi/dali/ |

## Strain construction

*Mycobacteria smegmatis* mc(2)155 (wild type) and ∆*ideR* cells were chromosomally tagged using the ORBIT protocol (*Figure 1—figure supplement 1*). For wild type cells, the integrase and annealase expressing plasmid was pKM444. For recombineering in the ∆*ideR* strain, which already contained a kanamycin resistance marker, we created a modified pKM444 plasmid with a zeocin resistance cassette inserted at the EcoIV restriction site. The tagging plasmid was pKM468 with a 3C protease cleavage site added before the EGFP tag. The targeting oligo had the sequence: 5' TGTGCG TTCCACTGGTTCCCCGGCAACCACCTGCTGCACGTGAGCCAGCCGGACTACCTAGGTTTGTACCG

TACACCACTGAGACCGCGGTGGTTGACCAGACAAACCCGCCGGATGACCCGCTTCC TGCGCGGCTTCATGTTCGACTGAACCCTTCACCGAGGTCCG 3'. *M. smegmatis* cells containing pKM444 were grown in an overnight liquid culture and induced for annealase and integrase expression. Cells were prepared for electroporation and electroporated with the targeting oligo and tagging plasmid. The transformed *M. smegmatis* were plated on hygromycin (wild type) or hygromycin and kanamycin (ΔideR) containing 7H9 plates and incubated at 37° C for 3 days. Colonies were verified for insertion of the tagging plasmid into the chromosome by PCR.

## Western blotting

100 mL of EccE$_3$ tagged wild type and ΔideR knock out cells were grown overnight to an OD$_{600}$ of 1.0–1.2. Cells were pelleted and resuspended in 1 mL of buffer (50 mM Tris-HCl pH 8.0, 150 mM NaCl, 1% DDM) and sonicated for 30 s. Cell lysates were run on a 4–20% SDS-PAGE gel (GenScript) and transferred to PVDF membrane (BioRad) using a BioRad Trans-Blot Turbo Transfer System. The blot was washed with PBS and blocked in a 5% milk/PBS-T solution for 1 hr. The blot was incubated with mouse anti-GFP monoclonal antibody (Roche) overnight. After rinsing with PBS-T, the blot was incubated with anti-mouse IgG HRP-conjugated antibody (R&D Systems) for 2 hr. After activation (Amersham) the blot was imaged on a BioRad ChemiDoc. The blot was stripped with stripping buffer (ThermoFisher Scientific) as per the manufacture's instructions, and incubated overnight with rabbit anti-GroEL monoclonal antibody (Sigma-Aldrich). The blot was incubated with goat anti-rabbit IgG antibody HRP (GenScript) for 2 hr, activated (Amersham), and imaged on a BioRad ChemiDoc.

## Protein purification

Purification for high resolution structural determination: *M. smegmatis* was grown in 6 L of 7H9 supplemented with 0.05% Tween 80 and 20 μg/mL kanamycin to an OD$_{600}$ of ~0.8. After harvest, cells were washed three times with PBS and frozen in liquid nitrogen before lysis with a cryogenic grinder (SPEX SamplePrep). 24.9 g of powdered cell material was resuspended by adding 56.3 mL 50 mM Tris-HCl pH 8.0, 150 mM NaCl, 1% DDM supplemented with 1X protease inhibitor cocktail (Sigma-Fast) and 224 units Benzonase endonuclease. The suspension was stirred for 120 min at 4°C. After centrifugation for 30 min at 98,000 g, the supernatant was incubated with 1.4 mL anti-GFP-nanobody resin for 110 min at 4°C. The resin was transferred to a column and washed sequentially with 28 ml of wash buffer (50 mM Tris-HCl pH 8.0, 150 mM NaCl and 0.1% GDN), 14 mL of high salt wash buffer (50 mM Tris-HCl pH 8.0, 400 mM NaCl, and 0.1% GDN), and 14 mL of wash buffer (50 mM Tris-HCl pH 8.0, 150 mM NaCl, and 0.1% GDN). To cleave off the purification tag, the resin was incubated o/n at 4°C with 70 units Pierce HRV 3C protease (Thermo Scientific Pierce) in 2.8 mL wash buffer supplemented with 0.2 mM DTT. This resin was sedimented by gentle centrifugation (300 x g for 3 min), the supernatant collected, and the resin was subsequently washed with 1.4 mL wash buffer. The supernatant and wash fraction were combined and concentrated using an Amicon Ultra-4 centrifugal filter unit with a 100 kDa molecular weight cut-off. The sample was centrifuged at 16,000 g before injection on a Superose 6 10/300 column equilibrated in 50 mM Tris-HCl pH 8.0, 150 mM NaCl and 0.021% GDN. Peak fractions were concentrated using a 0.5 mL centrifugal filter unit (Amicon, 100 kDa cut-off) to an A280 of 5.52 by Nanodrop reading in about ~30 μL . Purification completed for examination of the void fractions was similar except: volumes were scaled for a powder weight of 21.1 g. and the high salt wash was omitted.

## Blue-Native polyacrylamide gel electrophoresis (BN-PAGE)

BN-PAGE experiments were carried out using the Invitrogen NativePAGE Novex Bis-Tris Gel system as recommended by the manufacturer. Samples were prepared in a total volume of 10 μL using 0.5 μL 5% G-250 sample additive. Electrophoresis was performed at a constant voltage of 105-120 V for 2-3.5 hr at 4°C. The gel was fixed and stained using the Pierce silver stain kit.

## Cryo-EM – data acquisition

Samples were frozen for cryo-EM. Quantifoil R1.2/1.3, 400 mesh, copper grids were glow discharged using a Solarus plasma cleaner (Gatan) with an H$_2$/O$_2$ mixture for 30 s. 2 μL of sample were applied per grid and the grids were plunged into liquid ethane using a FEI Vitrobot Mark IV.

Initially, samples were screened, and test data sets were collected on a FEI Talos Arctica 200kV microscope equipped with a Gatan K2 Summit detector. For the initial screen of freezing conditions, 2499 movies were collected at a magnification of 36,000 with a pixel size of 1.14, and a defocus range of −1.5 to −2.5 μm, an exposure time of 9 s, and a dose rate of 7 electrons/Å$^2$/second (*Table 1*). Data collection for the final structure presented in the main text was collected on a FEI Titan Krios at 300kV with a Gatan K2 Summit detector. Two imaging sessions were used. In the first imaging session, 2705 movies were collected at a magnification of 29,000 with a pixel size of 0.82, and a defocus of −0.4 to −1.2 μm, an exposure time of 10 s to collect 100 total frames, and a dose rate of 8 electrons/Å$^2$/second (*Table 1*). In the second imaging session, data was collected on the same microscope with the same detector, 4632 movies were collected at a magnification of 29,000 with pixel size of 0.82, and a defocus range of −0.6 to −1.4 μm, an exposure time of 10 s to collect 80 total frames, and a dose rate of 6.7 electrons/Å$^2$/second. Data used to analyze the void, plateau, and peak regions of the SEC profile were collected on a FEI Talos Arctica at 200kV with a Gatan K3 detector. All micrographs were collected at a magnification of 28,000 with a pixel size of 0.9, and a defocus range of −1.5 to −2.5 μm, an exposure time of 11.7 s to collect 117 total frames at a total dose of 58 electrons/Å$^2$. For the void region, 1215 micrographs were collected.

## Cryo-EM – data processing

For all data, movies were motion corrected using MotionCor2 (*Zheng et al., 2017*) and CTF correction was performed using CTFfind4 (*Rohou and Grigorieff, 2015*). For the Arctica dataset, particles were picked using a gaussian blob in either RELION (*Zivanov et al., 2018*) or cisTEM (*Grant et al., 2018*) and initial 2D classification was performed to remove obvious artifactual particles. Initially, a shotgun approach was taken to generate several initial models using RELION, cisTEM, and cryosparc (*Punjani et al., 2017*). Once an initial model which contained realistic low-resolution features was generated, a user defined descent gradient was performed to improve the model with the goal of achieving accurate secondary structure features. First, all particles selected during 2D classification were refined in 3D against the randomly generated initial model. Second, a round of 3D classification with four classes and default RELION settings was performed and the best class selected. Third, the best class was refined as a single class in 3D classification with increasing Tau2_Fudge and decreasing search angle size. The resulting EM density map had clear transmembrane helix densities and was used as the model for a new 3D reconstruction. This reconstruction was used to back project models for reference-based particle picking in RELION. Two rounds of 2D classification were performed and the best classes selected. One round of 3D classification was performed using the Tau2_Fudge value optimized during the previous run through (T = 12) and the best class selected. A final 3D reconstruction of the Arctica data set yielded a map of about 4.7 Å resolution (*Figure 1— figure supplement 4*).

After motion correction and CTF determination, the final Titan Krios dataset was processed entirely using RELION. Particles were picked using a gaussian blob, and extracted as 4x binned particles. Two rounds of initial 2D classification were performed with T = 3 on the binned particles and obvious artifactual particles were removed. The final reconstruction from the Arctica dataset was used as the initial model for a 3D reconstruction of the binned particles. 3D classification with four classes and the previously optimized Tau2_Fudge value, T = 12, was performed on the binned particles. The two best classes were selected and re-extracted without binning. A 3D reconstruction was performed. A mask was created for the high-resolution region of the reconstruction and 3D classification without image alignment was performed focused on this region. The best class was selected and the subsequent 4.0 Å reconstruction is the consensus structure for the entire complex (*Figure 1—figure supplement 5*). Focused classification of each protomer, the periplasmic EccB region, and the ATPase 1, 2, and 3 domains of EccC were performed. To perform focused classification, the center of mass of the region of interest was determined using chimera (*Pettersen et al., 2004*). Particles were recentered on this area and reextracted. Masks for the region of interest were generated and 3D classification without image alignment was performed. The best class was selected and used for a focused 3D reconstruction without image alignment of the region of interest. A reconstruction was generated and density outside of the region of interest was subtracted. A final reconstruction of the masked and density subtracted particles was then performed. This procedure improved the resolution of the protomer i to 3.75 Å and protomer ii 3.83 Å, 5.8 Å resolution for the EccB$_3$ periplasmic domain, and ~7 Å resolution for the EccC$_3$ lower cytoplasmic region.

To generate the symmetry expanded protomers based on non-point group symmetry (also known as non-crystallographic symmetry or NCS), a transformation matrix between the two protomers was calculated using chimera. Particles were then transformed and aligned using the subparticles.py and star.py utilities in pyem (*Asarnow et al., 2019*) resulting in a particle stack with twice as many particles as the input file, each focused on protomer i or protomer ii. Density subtraction was performed to remove density outside of the symmetry expanded protomer, and focused classification and refinement were performed as described above. This procedure improved the resolution of the symmetry expanded protomer to 3.69 Å resolution.

## Atomic model building

The cytoplasmic domain from the crystal structure of $EccD_1$ (PDB 4KV2) was docked into the cytoplasmic domains of the two $EccD_3$ molecules and the sequence was mutated. The remaining transmembrane domains of $EccD_3$ and the residues 14–93 of $EccB_3$ were built *de novo* in Coot (*Emsley et al., 2010*) using baton building. The alpha helices of $EccE_3$ and $EccC_3$ were initially modeled using the RaptorX (*Källberg et al., 2012*) homology server. The loops and strands of $EccE_3$ and $EccC_3$ were built in Coot using baton building. All models were subsequently refined individually, as a symmetry expanded protomer, left and right protomers, and as the full model using phenix real space refine (*Afonine et al., 2018*), Coot, and the MDFF (*Trabuco et al., 2009*) server, Namdinator (*Kidmose et al., 2019*; *Supplementary file 1*).

## Low resolution modeling

The left and right protomer map, periplasmic focused refined map, and lower cytoplasmic focused refined map were all docked into the consensus map and added together using chimera. The combined map was filtered to 10 Å resolution to match the lowest resolution component. Homology models for amino acids 94–516 of $EccB_3$, the transmembrane helixes of $EccC_3$, and 404–1268 of $EccC_3$ were generated using RaptorX. These models were fit into the combined map density using the fit map to model utility in Chimera. The full model was refined using phenix.real_space_refine.

## Model interpretation and display

Buried surface area between subunits was calculated by PISA (*Krissinel, 2015*). Atomic models for individual proteins were compared against the PDB using the DALI server (*Holm and Laakso, 2016*). Chimera and ChimeraX (*Goddard et al., 2018*) were used to display maps and models for figure creation. Consurf (*Ashkenazy et al., 2016*) was used to produce multisequence alignments and to color structural models by homology.

# Acknowledgements

The ΔideR strain was provided by Gloria Marcela Rodriguez. Data was collected at the electron microscopy core facility at the University of California, San Francisco with the assistance of Alexander G Myasnikov. We thank Daniel Asarnow for making his pyem software available on github ahead of publication. We thank Huong Kratochvil and William DeGrado for preliminary lipid modeling and helpful discussions. We thank Robert Stroud and the members of his laboratory for stimulating discussions. We have used resources from NIH grants S10OD020054 and S10OD021741.

# Additional information

### Funding

| Funder | Grant reference number | Author |
| --- | --- | --- |
| National Institutes of Health | 1RO1AI128214 | Oren S Rosenberg |
| National Institutes of Health | 1U19AI135990-01 | Oren S Rosenberg |
| National Institutes of Health | P01AI095208 | Oren S Rosenberg |
| National Institutes of Health | 5T32AI060537 | Nicole Poweleit |

The funders had no role in study design, data collection and interpretation, or the decision to submit the work for publication.

## Author contributions
Nicole Poweleit, Investigation, Formal analysis, Validation, Visualization, Methodology, Writing – Original Draft Preparation, Writing – Review & Editing; Rachel Nakagawa, Resources, Investigation, Methodology, Writing – Review & Editing; Donovan D Trinidad, Investigation, Methodology, Validation; Kenan C Murphy, Resources, Investigation, Methodology; Christopher M Sassetti, Resources, Validation, Methodology, Supervision, Writing – Review & Editing; Oren S Rosenberg, Conceptualization, Resources, Formal analysis, Supervision, Funding acquisition, Validation, Visualization, Methodology, Project administration

## Author ORCIDs
Nicole Poweleit https://orcid.org/0000-0002-2700-0241
Nadine Czudnochowski https://orcid.org/0000-0002-3146-4110
Donovan D Trinidad https://orcid.org/0000-0002-1439-9927
Oren S Rosenberg https://orcid.org/0000-0002-5736-4388

## Decision letter and Author response
Decision letter https://doi.org/10.7554/eLife.52983.sa1
Author response https://doi.org/10.7554/eLife.52983.sa2

# Additional files
## Supplementary files
• Supplementary file 1. Model Refinement Statistics.
• Supplementary file 2. Top Dali server hits.
• Supplementary file 3. Buried surface area.
• Transparent reporting form

## Data availability
The map files have been deposited at the EMDB with code 20820. The entry is online at https://www.ebi.ac.uk/pdbe/entry/emdb/EMD-20820. The model has been deposited at the PDB with the code 6UMM. It is online at http://www.rcsb.org/structure/6UMM.

The following datasets were generated:

| Author(s) | Year | Dataset title | Dataset URL | Database and Identifier |
|---|---|---|---|---|
| Poweleit N, Rosenberg OS | 2019 | A complete structure of the ESX-3 translocon complex | https://www.rcsb.org/structure/6UMM | RCSB Protein Data Bank, 6UMM |
| Poweleit N, Rosenberg OS | 2019 | A complete structure of the ESX-3 translocon complex | https://www.ebi.ac.uk/pdbe/entry/emdb/EMD-20820 | Electron Microscopy Data Bank, 20820 |

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
