## [Decision Letter]

**Acceptance summary:**

This manuscript by Poweleit et al. describes the cryoEM structure of a natively purified ESX-3 (Type VII) secretion system from *Mycobacterium smegmatis*. Our understanding of the type VII secretion mechanism has been limited by the lack of high-resolution structural information of the fully assembled, ~900 kDa machine. Here the authors isolate the native ESX-3 complex from *M. smegmatis* and solve the structure by single-particle cryoEM in detergent. This is a beautiful and informative structure, and the analysis is rigorous. The work is significant because it corroborates the structural observations recently published by a competing group (Famelis et al., 2019), and provides a framework for understanding how this secretion machine transports its folded clients across the cytoplasmic membrane.

**Decision letter after peer review:**

Thank you for submitting your article "The native structure of an ESX translocon" for consideration by *eLife*. Your article has been reviewed by four peer reviewers, one of whom is a member of our Board of Reviewing Editors, and the evaluation has been overseen by Richard Aldrich as the Senior Editor. The reviewers have opted to remain anonymous.

The reviewers have discussed the reviews with one another and the Reviewing Editor has drafted this decision to help you prepare a revised submission.

Summary:

This manuscript by Poweleit et al. describes the high-resolution structure of an ESX-3 complex of *M. smegmatis* obtained by cryo-electron microscopy. A virtually identical structure of the same complex has been published in Nature by Famelis et al. as the authors were preparing to submit this paper. In contrast to the structure published by Famelis et al., which was obtained by overexpressing the ESX-3 system from a plasmid, Poweleit et al. tagged the complex chromosomally, on the last gene (EccE) of the operon, to bypass potential transcriptional defects. While the presented work therefore represents a native structure, as opposed to the already published structure by Famelis and colleagues, the observed structural features/composition are the same for both structures. It was felt that the *eLife* policy on "scoops" protects the authors of this manuscript and that it therefore should be considered further after suitable revisions.

Essential revisions:

1) The authors fail to see any higher order oligomers, which have been previously reported in native (Houben et al., 2012, van Winden et al., 2016) and also plasmid expressed conditions (Famelis et al., 2019; Beckham et al., 2017). These publications have reported higher order oligomers in multiple mycobacterial species (*M. bovis* BCG, *M. marinum* and *M. smegmatis*) and from different type VII secretion systems (ESX-1, ESX-3 and ESX-5). Higher order multimerization has also been suggested/observed by the corresponding author in previously published work (Rosenberg et al., 2015) for the EccC ATPase. Importantly, Beckham et al. and Famelis et al. showed that these higher order oligomers are only present in the void peak fraction of the SEC analysis. In the current study it is unfortunately not mentioned which SEC fractions were used for the cryo-EM analysis, but it is suspected that the void peak fractions were not analyzed. To confirm that the native ESX-3 complex forms higher order oligomers, the authors should more thoroughly investigate the presence of these multimers, e.g. by including the void peak fractions in the analysis and/or analyzing the ESX-3 membrane complex by BN-PAGE analysis before SEC (including crosslinking).

2) The authors propose an alternative and contentious model for protein translocation through the membrane complex, i.e. through a hydrophobic cavity formed by two EccD subunits, which is (partially) filled with lipids. Indeed, the presence of such a large cavity in between these two subunits is striking. However, the surface of this cavity consists, except for a number of polar residues, solely of hydrophobic residues. In agreement with a hydrophobic interface, the structure reveals extra densities at the periplasmic part of the cavity suggesting the presence of lipids or detergent. The cytoplasmic half of the cavity does not reveal these extra densities, which could be caused by the presence of several polar residues here, as also discussed by the authors. However, lipids in this part of the cavity could have been removed during solubilization of the complex. To show that partially lipid-filled cavities have been observed also in other membrane complexes, the authors refer to a publication describing the presence of a similar lipid plug in the central cavity of a reconstituted rotor cylinder of the ATP synthase (Meiers et al. Febs Lett 2001). However, it is explicitly mentioned in this article that "As the detergent-purified c cylinder is completely devoid of phospholipids, these are incorporated into the central hole from one side of the cylinder during the reconstitution procedure". It therefore remains unclear whether such lipid-plugged membrane pores actually exists in vivo. In addition, Poweleit et al. propose a "lipid mediated" mechanism of protein translocation through this hydrophobic pore. It is difficult to envision though how lipids would be able to mediate translocation of folded substrate dimers with mostly hydrophilic surfaces. This would be a completely new mechanism of protein translocation not seen for any other protein secretion machineries. In addition, the authors speculate even further that besides (hydrophilic) proteins also hydrophobic molecules, such as lipids, could be transported through this cavity, while there is no clear evidence that type VII secretion systems are able mediate lipid transport. In conclusion, while it is indeed striking that such a large cavity is present in the ESX-3 complex, the authors should more cautiously discuss this model, by also mentioning the major problems with this model as discussed above. The translocation model of substrates through the central pore of a hexameric complex is far less speculative, as it is supported by experimental data, published by several groups (as also mentioned in the Discussion). This model should therefore get more emphasis.

3) The regulatory role of the "cytoplasmic bridge" is speculative and should be more carefully discussed. e.g. "multimeric cytoplasmic domain is poised to regulate the translocation of ESX substrates" and "EccC_3_ and EccE_3_ collaborate to buttress the cytoplasmic domains of EccD_3_ and form a stable regulatory bridge".

---

## [Author Response]

Essential revisions:1) The authors fail to see any higher order oligomers, which have been previously reported in native (Houben et al., 2012, van Winden et al., 2016) and also plasmid expressed conditions (Famelis et al., 2019; Beckham et al., 2017). These publications have reported higher order oligomers in multiple mycobacterial species (M. bovis BCG, M. marinum and M. smegmatis) and from different type VII secretion systems (ESX-1, ESX-3 and ESX-5). Higher order multimerization has also been suggested/observed by the corresponding author in previously published work (Rosenberg et al., 2015) for the EccC ATPase.

We simply do not have definitive proof of the “physiological” multimeric state of the protein complex and so decided not to speculate extensively about this important issue in our first submission. Prior work, well characterized in the editor’s review, and including our published work, has strongly suggested higher order structural and functional oligomerization. In particular, the requirement for a an “R-finger” mechanism for EccC ATPase activity – a hallmark of multimer formation – makes it clear that at some point in the secretion cycle, the ATPase 1 domains act as oligomers. In the structure we present in this manuscript the ATPase 1 domains, which have been presumed to be the active motor domains based on structural, biochemical and genetic data, would require a total of 65 Å of movement to come into an active position where the R-finger from one protomer is positioned to stimulate catalysis in the next subunit. The requirement for this extraordinarily large movement of the ATPase domains makes it difficult to presuppose the path of the ATPase domains during the secretion cycle. In our revision we comment on this issue in the Discussion and alter the model figure (revised Figure 6) to reflect the need for these molecular motions.

Importantly, Beckham et al. and Famelis et al. showed that these higher order oligomers are only present in the void peak fraction of the SEC analysis. In the current study it is unfortunately not mentioned which SEC fractions were used for the cryo-EM analysis, but it is suspected that the void peak fractions were not analyzed. To confirm that the native ESX-3 complex forms higher order oligomers, the authors should more thoroughly investigate the presence of these multimers, e.g. by including the void peak fractions in the analysis and/or analyzing the ESX-3 membrane complex by BN-PAGE analysis before SEC (including crosslinking).

In order to address this important concern, we have repeated the purification and collected a new cryo-EM data set of of the “void” fractions. These micrographs contain a majority of aggregated particles and are challenging to analyze, which is why we left them out of the first submission. However, there is a small population of monodispersed particles, which we have analyzed. The majority of these particles appear to be in the dimer state we have described, however, additionally, we see a small set of side views that are consistent with a larger particle of approximately 280 Å and that look similar to particle described in Famelis et al. In terms of the size, these particles could be interpreted as a hexameric form, but given the excellent distribution of views we observe in the dimer form we were surprised not to observe any evidence in the 2D class averages that would suggest symmetry higher than 2. It is certainly possible that these particles do not represent a fixed state but rather misincorporation of several dimers into a single micelle. We present these data in a new supplemental figure (Figure 1—figure supplement 4).

We have also attempted to crosslink the purified protein before running on the SEC and we do not see evidence of additional, significant crosslinking between dimers, in the purified preparation. We include this gel in Author response image 1 but not in the resubmitted manuscript.

2) The authors propose an alternative and contentious model for protein translocation through the membrane complex, i.e. through a hydrophobic cavity formed by two EccD subunits, which is (partially) filled with lipids. Indeed, the presence of such a large cavity in between these two subunits is striking. However, the surface of this cavity consists, except for a number of polar residues, solely of hydrophobic residues. In agreement with a hydrophobic interface, the structure reveals extra densities at the periplasmic part of the cavity suggesting the presence of lipids or detergent. The cytoplasmic half of the cavity does not reveal these extra densities, which could be caused by the presence of several polar residues here, as also discussed by the authors. However, lipids in this part of the cavity could have been removed during solubilization of the complex.

As the reviewers note, the presence of a large cavity in the inner membrane, formed by the EccD dimer, is very unusual. In searching the literature and doing extensive homology searches, we have discovered no clear examples of similar membrane protein structures. In our structure, the periplasmic side of the EccD cavity is completely filled with detergent or lipids carried over from the purification, which would provide a mechanism for the “hole” to be sealed. Although we do believe that this cavity will eventually be shown to be of functional importance, as the reviewers appear to recognize, to fully understand the native state and function of EccD is beyond the scope of the current work. We have thus rewritten the Results and Discussion section to be more descriptive and less speculative.

To show that partially lipid-filled cavities have been observed also in other membrane complexes, the authors refer to a publication describing the presence of a similar lipid plug in the central cavity of a reconstituted rotor cylinder of the ATP synthase (Meiers et al. Febs Lett 2001). However, it is explicitly mentioned in this article that "As the detergent-purified c cylinder is completely devoid of phospholipids, these are incorporated into the central hole from one side of the cylinder during the reconstitution procedure". It therefore remains unclear whether such lipid-plugged membrane pores actually exists in vivo.

We have removed the reference to the c cylinder as we do not want to confuse the reader with a loose analogy that may not provide useful evidence.

In addition, Poweleit et al. propose a "lipid mediated" mechanism of protein translocation through this hydrophobic pore. It is difficult to envision though how lipids would be able to mediate translocation of folded substrate dimers with mostly hydrophilic surfaces. This would be a completely new mechanism of protein translocation not seen for any other protein secretion machineries. In addition, the authors speculate even further that besides (hydrophilic) proteins also hydrophobic molecules, such as lipids, could be transported through this cavity, while there is no clear evidence that type VII secretion systems are able mediate lipid transport. In conclusion, while it is indeed striking that such a large cavity is present in the ESX-3 complex, the authors should more cautiously discuss this model, by also mentioning the major problems with this model as discussed above. The translocation model of substrates through the central pore of a hexameric complex is far less speculative, as it is supported by experimental data, published by several groups (as also mentioned in the Discussion). This model should therefore get more emphasis.

We have restructured and rewritten the Discussion to further emphasize the relationship between the prior literature and the oligomer model.

3) The regulatory role of the "cytoplasmic bridge" is speculative and should be more carefully discussed. e.g. "multimeric cytoplasmic domain is poised to regulate the translocation of ESX substrates" and "EccC_3_ and EccE_3_ collaborate to buttress the cytoplasmic domains of EccD_3_ and form a stable regulatory bridge".

Our intension with this language was to point out the interactions between the cytoplasmic domain of EccD_3_ and the other two proteins with significant cytoplasmic domains, EccE_3_ and EccC_3_. EccD_3_ contains a highly conserved region (116-125) in the linker connecting the ubiquitin-like domain at the N-terminus to the transmembrane region. Despite the strong conservation, the interface made by this region each copy of EccD_3_ in the homodimer is strikingly different. In one case, the region interacts extensively with the other cytoplasmic domain of EccD_3_ as well as with EccE, anchoring EccE in a rigid conformation. In contrast, this region has a completely different conformation in the symmetry related copy of EccD, where it reaches in a different direction to form a nexus of interactions with EccB, and EccC. Given the need for the rearrangement of EccC to accommodate the R-finger mechanism of the ATPase 1 domain, we speculate that flexibility of the EccD_3_ linker could allow for the release of EccE and EccC from their rigid positions, thus allowing for a rearrangement of the ATPase domains into an active conformation. To respond to the comments. we have clarified this idea of a “regulatory bridge” and separated the observations of the structure from the speculation about the functional relevance, which is now in the Discussion. We have updated and expanded Figures 3C-D, Figure 4C-D and Figure 5C to highlight these interactions.